# Developing an Animal Welfare Assessment Protocol for Cows in Extensive Beef Cow-Calf Systems in New Zealand. Part 2: Categorisation and Scoring of Welfare Assessment Measures

**DOI:** 10.3390/ani10091592

**Published:** 2020-09-07

**Authors:** Y. Baby Kaurivi, Rebecca Hickson, Richard Laven, Tim Parkinson, Kevin Stafford

**Affiliations:** 1School of Veterinary Medicine, Massey University, Private Bag 11 222, Palmerston North 4442, New Zealand; r.laven@massey.ac.nz (R.L.); t.j.parkinson@massey.ac.nz (T.P.); 2School of Agriculture and Environmental Management, Massey University, Private Bag 11 222, Palmerston North 4442, New Zealand; R.Hickson@massey.ac.nz (R.H.); k.j.stafford@massey.ac.nz (K.S.)

**Keywords:** categorisation, welfare assessment, extensive beef cows, New Zealand

## Abstract

**Simple Summary:**

Animal welfare assessment protocols use different methods to categorise and score animal welfare. This study has demonstrated the feasibility of developing standards for a welfare assessment protocol of cow-calf farms in New Zealand by validating potential categorisation thresholds for measures of assessment on 25 beef farms. Imposed thresholds of categorisation and derived thresholds based upon the poorest 15% and best 50% of farms for each measure were compared to see which was the most appropriate to the range of observations and the significance of the welfare implications of the measure. For measures with significant welfare implications, the stricter threshold was retained, while derived thresholds appeared more appropriate for commonly occurring traits but of less welfare importance for the production system at hand.

**Abstract:**

The intention of this study was to develop standards for a welfare assessment protocol by validating potential categorisation thresholds for the assessment of beef farms in New Zealand. Thirty-two measures, based on the Welfare Quality and the University of California (UC) Davis Cow-Calf protocols, plus some indicators specific to New Zealand, that were assessed during routine yardings of 3366 cattle on 25 cow-calf beef farms in the Waikato region were categorised on a three-point welfare score, where 0 denotes good welfare, 1 marginal welfare, and 2 poor/unacceptable welfare. Initial categorisation of welfare thresholds was based upon the authors’ perception of acceptable welfare standards and the consensus of the literature, with subsequent derived thresholds being based upon the poorest 15% and best 50% of farms for each measure. Imposed thresholds for lameness, dystocia, and mortality rate were retained in view of the significance of these conditions for the welfare of affected cattle, while higher derived thresholds appeared more appropriate for dirtiness and faecal staining which were thought to have less significant welfare implications for cattle on pasture. Fearful/agitated and running behaviours were above expectations, probably due to the infrequent yarding of cows, and thus the derived thresholds were thought to be more appropriate. These thresholds provide indicators to farmers and farm advisors regarding the levels at which intervention and remediation is required for a range of welfare measures.

## 1. Introduction

An on-farm assessment of animal welfare requires a summation of measures which indicate the overall welfare status of farms. One key element of animal welfare assessment is the benchmarking of welfare measures across similar production operations and a summation of these measures [1]. Benchmarking farm animal welfare can be used to identify welfare problems and remediate them, and to allow follow-up assessments [2,3,4].

There is no simple way of measuring animal welfare [5,6,7] and grading animal welfare compromise is not easy [1,8,9]. The consensus for providing an overall judgement of animal welfare on a farm where different welfare measures are aggregated and weighted [8,10,11], is still contested [1,9]. The controversy emanates from the challenging interpretation of weighted sums that may tolerate compensation between welfare principles, such as an inappropriate environment being countered by good health [8,11]. Attempts to remove all compensation between welfare aspects in the Welfare Quality Assessment [12] were not achieved [11,13] and aggregation in this system was criticised as illogically allowing the cover up of important welfare problems [1]. 

In most welfare assessment protocols data are collected in various scales. The expression of data as ordinal scores makes the summation into overall scores difficult [8,9,14]. In the Welfare Quality protocol, data collected from the farms are integrated in a sequential transformation and aggregation, where the final welfare score result is presented on a scale of 0–100, ranking farms as excellent, improved, acceptable, and not classified [8,15]. This aggregation of scores is a complex process, and it has been criticised as not being in line with expert opinion [11,15]. 

The aggregation of welfare assessments may impact animal welfare decisions when schemes are applied on-farm. Indeed, the goal of achieving a transparent, yet practicable-to-compute system [1], may be unattainable. Hence, the splitting of measures into classes differing in severity has been recommended to give meaningful interpretation of scores on-farms [8]. Grading in the Five Domain Model comprises a five-tier scale grading from ‘no compromise’ to ‘extreme compromise’ [9]. The same author however, consented to the use of a simpler grading in cases of sparse or compromised data. The problem of having five-point grading systems on farm is that it is laborious and time consuming, which can reduce the feasibility of making such assessments [3,16]. This is even more impractical for extensive systems, or where animals are assessed during yarding where they may be moving fast in and out of the race with barely enough time to adequately assess each criterion. Hence, using an ordinal categorical score and computing the number of scores on a three-tier level (i.e., 0–2) might be an easier approach. A three-tier scale of severity was also endorsed by other authors [17,18]. 

Kaurivi [19] combined measures from the Welfare Quality protocol for beef cattle with the University of California (UC) Davis Cow-Calf Health and Handling protocol [20] and additional New Zealand-specific measures, to identify measures suitable for creating a welfare assessment protocol for extensively reared pasture-based beef cattle in New Zealand. Following the formulation phase, the protocol was then applied on 25 cow-calf farms in the Waikato region of New Zealand [21] to further test the protocol for completeness and on-farm feasibility. Through a process of eliminating unsuitable measures, adjustments of modifiable measures and retaining feasible measures, a protocol with 32 measures was created. However, the study did not attempt to categorise or rank welfare scores and welfare status of farms. The present study used the data collected by Kaurivi [21] to evaluate the best method to categorise animal welfare measures into scores that indicate a threshold of acceptable and non-acceptable welfare standards, as part of the development of an animal welfare assessment protocol for extensive beef systems in New Zealand. This aim was achieved through the setting and comparing of imposed categorisation thresholds with derived thresholds to see, in a commercial setting, which was the most appropriate to the range of observations and the significance of the welfare implications of the measure.

## 2. Materials and Methods

### 2.1. Protocol Used

The protocol was developed and trialled by Kaurivi [19] to create a robust, achievable suite of 32 measures which were usable on pasture-based extensive cow-calf beef farms in New Zealand [21]. Briefly, the protocol involved assessing measures combined from the Welfare Quality cattle protocol and the UC Davis Cow-Calf protocol, with additional New Zealand-specific measures. The assessment was trialled on 25 mixed sheep and cow-calf farms over 2 visits, with the first during routine yarding for pregnancy testing and the second using a questionnaire and observations of the herd at pasture during winter. In the first visit, a total of 4956 cows were presented for pregnancy diagnosis, with yard observations made on 3366 animals (see [19] for herd details). Observations were made of cows in the race regarding body condition, rumen-fill, behaviour and physical health. Stockpersonship was evaluated as cows entered and exited the race by observing how cows were handled. The yard design and handling facilities were also evaluated for ease of handling of cows. In the second visit, a farm resource evaluation and a questionnaire guided assessment of health and management of the herd was undertaken. 

### 2.2. Categorisation of Measures

Categorisation of scores was based on the authors’ experience and perception of what good welfare would be in extensive pasture-based beef cattle in New Zealand, together with a consensus developed from relevant literature. For each farm, the welfare impact of each of the 32 measures was categorised separately into 3 categories (Table 1). All discrete data were measured according to the proportion of cases and given an ordinal score of welfare on that 3-point category score. For example, the mean percentages of poor body condition (BCS; thin cows) and poor rumen fill (RFS; hungry cows) were given an ordinal score to indicate acceptable and unacceptable welfare in the herds. 

For consistency, thresholds of measures across welfare principles were kept constant where reasonable. For example, for good health, absence of injuries or physical impairment welfare categories were kept the same at 2% threshold. Painful conditions (dystocia and short tail) were also given a similar category to the health issues. Ordinal data (i.e., age of castration) and subjective measures (i.e., handlers’ noise) were similarly given a categorical score to indicate severity of marginal and unacceptable welfare in the herds. Finally, the categorisation of measures such as yarding frequency and health check frequency were moderated with respect to findings during the assessment visits. The categorisation and details on how each of these 32 measures were assessed are summarised in Table 2, Table 3 and Table 4.

### 2.3. Data Analysis

Data were analysed using SPSS version 24 (IBM). Descriptive statistics for continuous measures were used to capture central tendency (mean and median), dispersion of data (standard deviation), range (minimum and maximum), variance and percentiles. Qualitative methods were used to analyse the frequency of ordinal measures. The Shapiro–Wilk test was used to test for normality, and log10(n + 1) was used to transform those variables that were not normally distributed. An alternative approach to applying pre-determined value judgements is to determine the threshold from the data, so that an arbitrary 15% of farms were considered poor and 50% good derived thresholds were determined based on z scores to result in approximately 50% of farms falling into a good welfare band (‘green’) and 15% of farms into a poor welfare band (‘red’). Farms not in the green or red band were classified as orange. The arbitrary 15% was chosen to fit with the ‘15% rule’ where animals (in this case farms) below this point are considered as worse-off in terms of animal welfare compromise [12,18]. Farms were not given an ‘overall’ welfare score [9]. For each non-categorical measure, the derived red threshold and the imposed score 2 threshold were then compared by dividing the derived threshold by the imposed threshold.

## 3. Results

### 3.1. Welfare Assessments Summary Statistics 

Consolidated data for the welfare observations made using the final protocol are shown in Table 5 and Table 6. 

For painful management procedures, castration was performed with rubber rings on 20 of the 25 farms (mode and median two months of age; range 1–4 months). Calves were disbudded only on two farms, at three and four months respectively. Ear-tagging was performed without the use of anaesthesia on all farms at median and mode of two months. None of the farms reported high levels of diseases in the last 12 months (based on 2017 herd size). Of the diseases reported, only lameness (which was reported on 18 out of 25 farms), abortion (13 out of 25 farms), and dystocia (21 out of 25 farms) had a mean recorded incidence across all farms of greater than 1% (1.1%, 1.5%, and 2.6%, respectively). On individual farms, the highest recorded incidences of lameness, abortion and dystocia per herd were 4.1% (five out of 155 cattle), 8.8% (20 out of 226 and seven out of 80 cows) and 16.7% (two out of 12 cows), respectively. The mean incidence of all other diseases was < 0.5%, and of those diseases, only eye cancer (three out of 125; 2.4%), theileriosis (10 out of 470; 2.1%), vaginal prolapse (one out of 35; 2.9%), and Mg deficiency (10 out of 470; 2.1%) had maximum recorded incidences in an individual herd of >2%. (See Appendix A for the main diseases recorded as per farmers’ recollection in the questionnaire assessment at the 25 Waikato beef farms).

### 3.2. Categorisation of Measures

Categorised observational data are illustrated in Figure 1 (measures of feeding and environmental factors), Figure 2 (health determinations and frequencies of painful procedures), and Figure 3 (stockpersonship scores).

Most farms (17 out of 25) scored poorly for rumen fill, whereas no farms had poor welfare for distance to water. Poor welfare scores for short tails and dirtiness were reported at 14 out of 25 farms, whilst 22 out of 25 farms had faecal soiling (‘diarrhoea’). All farms except one obtained good welfare score for shade and no farms had a good score for environmental hazards. Most farms scored poorly for mortality rate followed by lameness. No farm was scored as having poor welfare associated with hair loss or abrasions. Only two farms disbudded calves and both did so after two months, which gave a poor welfare score. The rest had calves that were genetically polled. A poor welfare score was noted at five out of 25 farms for castration, and all farms received a marginal welfare score for ear tagging (tagging without local anaesthesia). No farm scored poorly for mis-catching, hitting cows and handling noise whereas 10 and 11 out of 25 farms were placed in category 2, for equipment noise and dog noise, respectively. Most yarding frequency scores were in category 1, and cattle were generally fearful and agitated in the yards.

The accumulated welfare score according to the 3-point scores for each farm is shown in Figure 4, with farms ordered from most poor scores to fewest (range 14–3). The highest number of good scores was 22 out of 32 and lowest was six. For marginal scores, the range was 6–17.

### 3.3. Refined Thresholds

Derived threshold values are shown in Table 7. Measures that were normally distributed were hungry cows, dirtiness, diarrhoea (faecal soiling), mortality rate and fearful/ agitated cows. Seven measures had a derived red threshold that was >2 times the threshold imposed by categorisation: short tail, diarrhoea, lameness, dystocia, mortality rate, fearful/ agitated, and cows running on exit. 

## 4. Discussion

Kaurivi [21] identified 32 measures of animal welfare that were feasible to assess during routine yarding of pasture-based beef cattle. This study categorised these animal welfare measures into scores that indicate a threshold of acceptable and unacceptable welfare, to provide guidance for when intervention was needed [8]. The thresholds that have been imposed or derived in this study are based on individual measures, rather than an aggregated ‘score’ for each farm [9]. 

Cattle were in good body condition at the time of assessment, with an average of 10.7% of cows having a BCS ≤ 4. The range of thin cows across farms was wide (0–60.7%). The imposed threshold for categorisation as poor welfare was 10% of the herd, whilst the derived threshold, based upon the poorest 15% of farms, was 19% of thin cows. In terms of identifying the need for nutritional intervention, the lower threshold seemed more appropriate, even though cows’ productivity is not impaired in the short term, at BCS 4 [23]. Other studies have suggested that a threshold for the proportion of thin cows that is deemed unacceptable could be set at 5–15% [24] and 6.7% [16]. The BCS data in the present study were largely correlated with rumen fill score (RFS) data, although, whilst a poor RFS can reflect long-term underfeeding it can also occur during short term feed deprivation [25], such as when cows are drafted a day before pregnancy testing. Hence, the derived threshold of ≤19% of the herd with a low RFS may not be more appropriate for the detection of poor nutrition than the original imposed figure of 50%. 

Assessing the dirtiness of cattle was both difficult and unrewarding. Kaurivi [21] concluded that all sites of dirt (tail, hindquarters and flank) should be amalgamated to provide a single ‘dirtiness’ score. They also noted the confounding of faecal staining of the tail head as a sign of infectious/parasitic diarrhoea with its very common occurrence in normal cattle that are being fed lush pasture. Hence, whilst in housed cattle, dirtiness and diarrhoea are rightly interpreted as signs of poor housing and/or health control, these interpretations may not be relevant to the study population. Rather, dirtiness and diarrhoea probably reflect the degree of muddiness of the paddocks and/or the lushness of the pasture which, in turn, are largely dependent on the season of the year. A point may be reached when the level of dirt in a pasture-based system does represents a welfare compromise [26], so creating standards for interpretation of dirtiness is therefore difficult [9,27]. The interpretation of dirtiness as a measure of welfare might require the setting of seasonal thresholds, e.g., finding dirty cows in January (summer lush pasture) is different to finding dirty cows in July (winter muddy terrain). Taken together, such considerations suggest that the derived threshold for red score of 36% of the herd being dirty seems more appropriate than the original imposed threshold of 20%. Likewise, the ubiquitousness of faecal staining due to the fluid nature of the cows’ normal faeces means that the derived threshold of close to 60% is probably more realistic than the imposed threshold of 20%. It could in fact be argued that, whatever threshold is used for faecal staining, it may represent the imposition of a characterisation of a trait that is poorly related to welfare compromise; rather, it is merely a sign of cows having plenty of grass. On the other hand, faecal soiling may contribute to a risk of disease [25], so perhaps adopting the re-categorised threshold of >60% (or 50%) may indeed provide a meaningful measure of welfare. Perhaps a qualitative determination may also need to be made of whether watery faeces are simply the result of the pasture diet or whether some identifiable disease process is causing abnormally loose faeces. Finally, there are economic implications associated with dirtiness in cases of cattle destined for slaughter [28] so, again, the scoring that is imposed might vary with the circumstances and/or purpose for which it is being undertaken. 

Assessing the incidence of short tails may help to determine whether faeces on tails does, or does not, represent compromised welfare, given that the aetiology of short tails is, in most cases, constriction of blood supply to the tails by hardened faecal rings. Short tails were present in 4.6% of cows, which compares unfavourably with the imposed standard of >2% of affected cows representing poor welfare. It seems reasonable to assume that the condition is associated with a significant level of pain to the cow, probably like that associated with tail docking with a rubber ring [29]. This is a good example of setting thresholds based on what should be achieved on-farm and not based on the status quo, and the envisioned scale could be used as a tool to caution farmers about the state of tail soiling, so that remedial actions can be taken to curb or prevent the occurrence of this condition (i.e., washing off the dirt or clearing the hardened faecal balls before the tail sloughs or breaks off). 

No farm in extensive hill or high-country in New Zealand is without any risk of hazards (i.e., steep hills, cliffs, streams, gullies and tomos). Farms (*n* = 8) that lost animals in tomos were considered as having major welfare compromise without considering the presence of the other hazardous terrains. Otherwise, the ranking of this measure was influenced by the prevailing conditions of the beef farms. Potential threats of the environment can never be eliminated, thus the application of strategies to minimise or bring accidents to tolerable levels would be more achievable [9]. The issue might be controlling the access of cows to these hazards rather than the presence of these hazards, so linking welfare compromise to good environmental management, such as preventing access to hazards, could provide a useful focus for reducing accidental death.

For most health-related measures, the welfare impacts were small on most farms. The exceptions were lameness, dystocia and mortality rate, for which the derived threshold was more than twice the imposed threshold. Importantly, whilst relatively low incidences of these conditions probably have relatively limited impact upon herd productivity per se, they have a very significant impact upon the individual cow. Thus, lameness is a critical welfare compromise indicator, as it is both a painful condition, and affects productivity [30,31]. The mean incidence of lameness in this survey was 2.7% (range 0–11.5%), but the incidence on the worst 15% of farms (derived threshold) was ≥4.8%, indicating that it has the potential to be a significant welfare issue. Consequently, the original imposed threshold of 2% is probably more appropriate than the derived threshold: particularly as the lower threshold would have the benefit of increasing the awareness of farmers to the need for intervention [31]. Whether it is appropriate to use a single ‘catch all’ criterion for lameness might be questioned [1,11]. For example, should lameness be differentiated into severe (non-weight-bearing) and non-severe lameness, with thresholds of 1% and 3% of the herd, respectively. On the other hand, in the circumstances in which observations were made in the present study, it was probably more accurate to use the catch all than to try to differentiate between levels of degrees of lameness. Dystocia similarly has a very significant impact upon the welfare of animals affected (and upon calves born/stillborn as a result of dystocia), and, at high incidences, can markedly impair the productivity of the farm [5]. Although the mean incidence, 2.6%, was close to the imposed limit, the derived limit was 4.9%, which indicates that dystocia is probably a relatively common trait on the beef farms. Again, given the significance of the condition for affected individuals, the 2% threshold seems more appropriate than the derived 5% limit. A threshold of 2% could aid in benchmarking for monitoring and correction of this condition. Finally, the average mortality rate was 3.9%, which is rather higher than the New Zealand industry standard for beef cattle (2–3% [32]). Similar figures have also been reported by international studies of pasture-based cow-calf units [33,34,35]. The threshold for the worst 15% of farms was 6.3%: given that mortality represents the total economic loss of the cow, mortality has the potential to be both common and economically serious on beef farms. Hickson [36] found a death rate of 2.1% per year in New Zealand beef herds, which is close to the imposed 2% threshold of the present study. Therefore, the 2% categorisation threshold appears to be a rational figure to trigger investigation of underlying contributing factors to reduce mortality rate.

The threshold for the ages above which performing painful management procedures (castration, removal of the horn bud) were considered as unacceptable welfare were set at >2 months. New regulations in New Zealand prohibit disbudding/dehorning without local anaesthesia, whilst the New Zealand Veterinary Association [37] advocates that these procedures should be undertaken at 2–6 weeks of age, and in conjunction with the use of analgesia [38]. For castration, this painful procedure can be mitigated using analgesia [39] and animals which are castrated early cope and recover faster than if this is done at an older age [39,40]. Ear notching is more painful than tagging, but the adverse effects can be mitigated using vapo-coolant [41] which provides a local cooling of the skin and thereby reduces pain perception. Hence, performing notching without the use of any anaesthetic was deemed to be a significant welfare compromise. 

Stockpersonship was categorised using ordinal measures related to the behaviour of the cattle in the yards and race, and categorical measures based upon observations of the stock handling. The ordinal measures ‘fearful/agitated’ and ‘run’ had derived thresholds that were more than twice the imposed threshold. Running was a common behaviour, for which the derived threshold (23.4%) was much higher than the imposed threshold (10%). Stumbling and falling were less common, with the derived and imposed thresholds being very similar at ~2%. Many of the stumbling cattle appeared to have been merely correcting their stance and hence might not warrant a stricter threshold. However, if extensiveness per se is the underlying cause, strategies such as more yarding events could be implemented to ascertain and prevent the welfare compromise. On the other hand, yarding is itself associated with stresses upon the cattle, so there are benefits to avoiding yarding cattle more often than is essential. In the present study, most farms (20/25) yarded the cattle 3–4 times per year, with the remainder of the farms yarding 5–6 times. A similar study of California ranches recorded an average of 3.4 yardings per year [42], but with a significant reduction in cattle vocalisation, stumbles and hitting with additional yardings per year [43]. This indicates an association of infrequent yarding and handling with difficult handling, restraining and fearfulness [44,45]. Concern around the infrequent yarding of cattle in extensive beef systems is supported by the finding in our study (Kaurivi Part 1) that yarding per year was correlated with fearful behaviour (ρ = 0.50). 

In the present study, the derived threshold for fearful/agitated behaviour was 4.9%, versus the imposed threshold of 2%. It is likely that the commonness of fearful/agitated behaviour may primarily be an indication of the lack of familiarity of extensively managed cattle with yarding and handling; as also found by Simon [43]. Taken together, it appears that the benefits of more frequent yarding (>4 times per year, for example) may be more compatible with acceptable welfare when cows are handled than yarding <3 times per year. Fortunately, the proportion of cows that were mis-caught during restraint when gates were closed into or within the race was low: even setting the threshold at 1% of cows mis-caught, only 4/25 farms exceeded that threshold. One explanation is that beef farmers do not routinely use a head bail for mass management procedures including pregnancy diagnosis (except on 8 out of 25 farms where the first cow was caught in the single-file race). Another explanation for this could be awareness of welfare compromise of this practice at New Zealand beef farms; a lower risk of mis-catching was reported if farmers undergo training in cattle handling techniques [43]. 

The frequency of health checks of cows by farmers during winter/pregnancy was based on the findings at the 25 beef farms, where health check frequency was regular with 20/25 farms inspecting at an interval of ≤1 week and 11/25 farms doing daily checks. Frequent health checks are expected to coincide with a good health status [43] and low mortality [46]. The limitation of health checks on extensive systems is that it is an overall inspection of cows at pasture, with rare close inspection to detect early health problems and injuries [47]. Thus, just recording health checks, without consideration of what health checks entailed while looking at individual cows versus the whole herd might influence the findings and hence categorisation of this measure [48]. 

This study was undertaken in only one region (Waikato) of New Zealand, thus the derived thresholds may reflect beef farming in that region rather than across the country. We concluded in our previous study that before the protocol was used widely it needed further testing on more farms across New Zealand with more assessors. This process should also include the calculation of derived thresholds for each of the measures in the protocol where data were collected on a continuous basis. These thresholds should, ideally, be calculated at a regional rather than a national level, so that, if present, differences between regions, can be highlighted. Only once this process is completed can we finalise the categorisation process started in this study. This finalisation process should involve beef farmers, beef exporters, animal welfare experts, veterinarians, consumers, and ideally, animal welfare advocacy groups.

## 5. Conclusions

This study has demonstrated the feasibility of developing standards for a welfare assessment protocol of cow-calf beef farms in New Zealand. Initial welfare thresholds were based upon the authors’ perception of acceptable welfare standards and the literature, with subsequent derivation of thresholds based upon the poorest 15% and best 50% of farms for each category. Imposed and derived thresholds were compared to see which was the most appropriate to the range of observations and the significance of the welfare implications of the measure. Some of the derived thresholds were much higher than those originally imposed, with lameness, dystocia, and mortality rate being between two and three times higher than the imposed threshold. Nonetheless, in view of the significance of these conditions for the welfare of affected cattle, the original threshold appeared the more appropriate. The proportion of cows with low BCS or RFS evoked similar considerations. On the other hand, measures of dirtiness and faecal staining were more common, but less significant than originally envisaged, so the derived thresholds appeared more appropriate. Similarly, measures of cow behaviour during handling were above expectations. Again, due to the infrequent yardings that these animals experienced, the derived threshold appeared to be the more appropriate. Taken together, these thresholds provide indicators to farmers and farm advisors regarding the levels at which intervention and remediation is required. Findings during the assessments that were supported by national and international standards also rationalised the categorisation of measures such as yarding frequency/year and health checks frequency. Further data are required from more assessments across the country in order to finalise the categorisation process started in this study.

## Figures and Tables

**Figure 1 animals-10-01592-f001:**
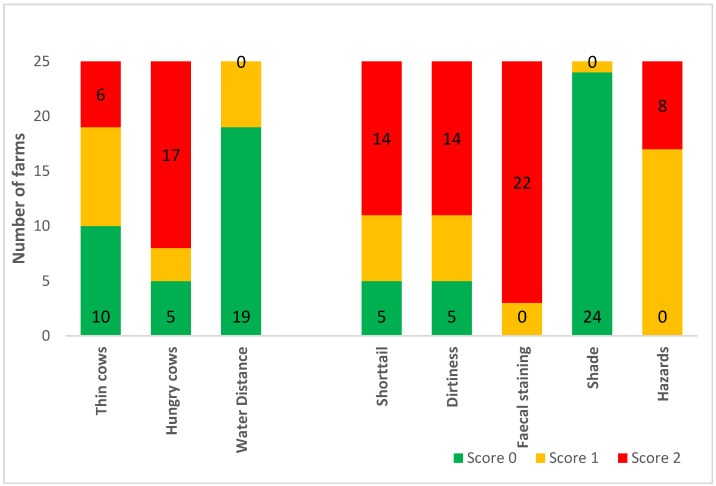
Frequency analysis of categorised good feeding and environment measures on the 25 Waikato beef farms, for which scores were assigned as either 0: good, 1: marginal, or 2: poor welfare. See Table 2 for further information on how each measure was categorised into a score of 0, 1 or 2.

**Figure 2 animals-10-01592-f002:**
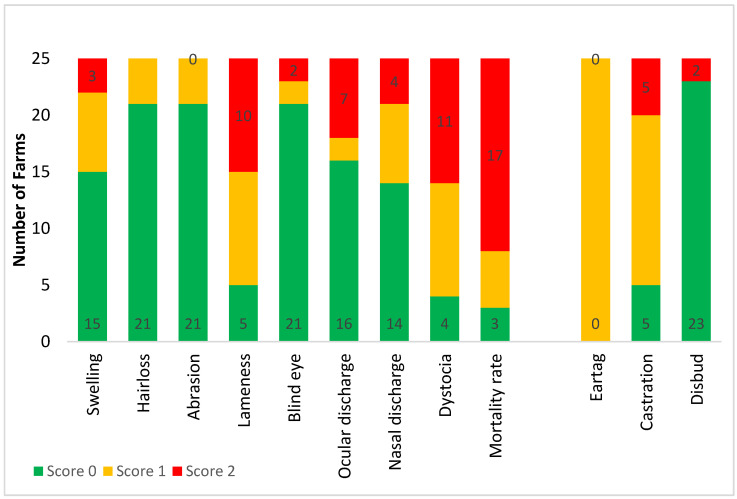
Frequency analysis of categorised good health measures on the 25 Waikato beef farms, for which scores were assigned as either 0: good, 1: marginal, or 2: poor welfare. See Table 3 for further information on how each measure was categorised into a score of 0, 1 or 2.

**Figure 3 animals-10-01592-f003:**
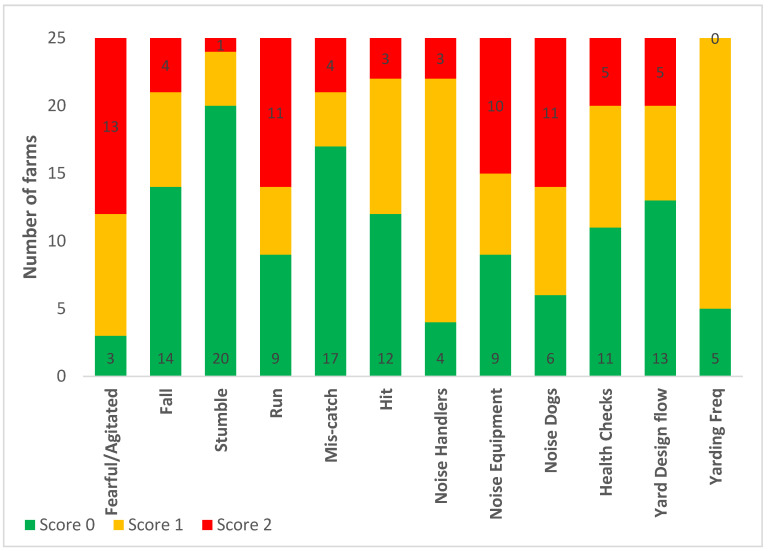
Frequency analysis of categorised appropriate stockpersonship measures on the 25 Waikato beef farms, for which scores were assigned as either 0: good, 1: marginal, or 2: poor welfare. See Table 2 for further information on how each measure was categorised into a score of 0, 1 or 2.

**Figure 4 animals-10-01592-f004:**
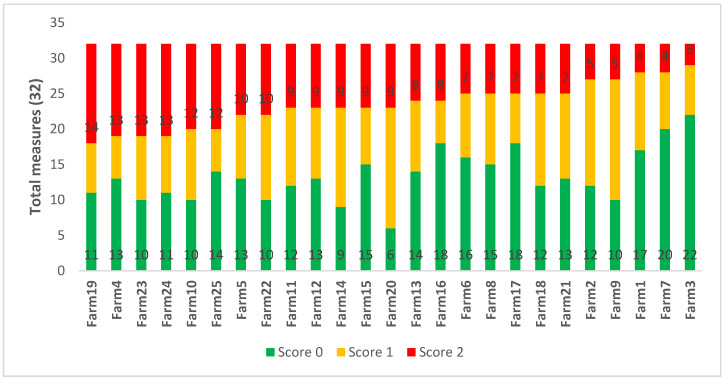
Accumulated categorised scores of the 32 measures according to the 3-point scores for the 25 Waikato beef farms, sorted from the most to the fewest score 2.

**Table 1 animals-10-01592-t001:** Categorisation of welfare scores.

Category Score	Welfare Assessment	Required Actions
0	Good/Acceptable	No intervention needed but keep monitoring
1	Marginal	Assess and plan for intervention. Increase monitoring
2	Poor/Unacceptable	Intervention needed immediately

**Table 2 animals-10-01592-t002:** Categorical ranking of good feeding and appropriate environment measures in the New Zealand cow-calf protocol.

Welfare Principles	Welfare Criteria	Animal Welfare Measure/Indicator	Scoring Description	Categorical Ranking
Good feeding	Absence of hunger	Body condition score (thin cows)	% thin in herd of score (score 1–4 on 1–10 scale; [22])	0: 0–5.0% 1: 5.1–10% 2: >10%
		Rumen fill score (hungry cows)	% of animals with hollow/ empty rumen observed in the race	0: 0–20.0% 1: 20.1–50% 2: >50%
	Absence of thirst	Distance and availability of water	Average distance to access water	0: 0–250.0 m 1: 250.1 ≤500 m 2: >500 m.
Appropriate environment	Comfort around resting	Short tail	% of observed cows with shortened tail	0: 0.0% 1: 0.1–2% 2: >2%
		Dirty body	Total number of animals assessed as having dirty tail, hind and flank	0: 0–10.0% 1: 10.1–20% 2: >20%
	Thermal comfort	Shade	Subjective assessment of shade in the paddocks (presence of trees, shrubs, galleys, man-made canopies)	0: sufficient 2: insufficient
	Ease of movement	Absence of hazardous objects/environment	Hazardous objects observed in the yard and paddocks (i.e., tomos *, sharp objects lying around)	0: no hazards 1: 1–2 hazards 2: 3 or more hazards or animals dying in any hazard)

* Tomos are underground caverns usually containing a watercourse, with small openings to the surface into which cattle can fall and die (animals dying in tomo was given a score 2 hazards).

**Table 3 animals-10-01592-t003:** Categorical ranking of health-related measures in the New Zealand cow-calf protocol.

Welfare Criteria	Animal Welfare Measure/Indicator	Scoring Description	Categorical Ranking in New Zealand Study
Absence of injuries/physical impairment	Swelling Hair loss/hairless Abrasions	% of observed cows with swelling, hairless patches or abrasions/fresh scratches (>1 cm)	0: 0.0% 8 81: 0.1–2% 2: >2%
Absence of disease	Lameness	% of observed cows with uneven weight weight-bearing on a limb that is immediately identifiable and/or obviously shortened strides	0: 0.0% 1: 0.1–2% 2: >2%
	Blindness Ocular discharges Nasal discharges	% of observed cows with ocular or nasal discharges extending 2cm, and those blind in one or both eyes	0: 0.0% 1: 0.1–2% 2: >2%
	Diarrhoea	% of observed cows with evidence of diarrhoea (more than a hand wide on both sides from base of tail)	0: 0–10.0% 1: 10.1–20% 2: >20%
	Dystocia	% of cows recorded with difficult births	0: 0.0% 1: 0.1–2% 2: >2%
	Mortality rate	% of accidental deaths, cattle which died due to disease, and those killed as a result of disease/accidents on the farm in the last 12 months	0: 0.0% 1: 0.1–2% 2: >2%
	Ear tagging/notching	Specify no tag or use of anaesthetics regardless of tagging or notching procedure and with/without the use of anaesthetic).	Ear tagging 0: no tag or use anaesthetics 1: tag with no anaesthetics 2: notching/cutting with no anaesthetics
Painful procedures	Castration Disbudding	Specify age at castration and use of anaesthetics Specify age at disbudding and use of anaesthetics	0: No castration/disbud 1: ≤2 months 2: >2 months

**Table 4 animals-10-01592-t004:** Categorical ranking of appropriate stockpersonship measures in the New Zealand cow-calf protocol.

Welfare Criteria	Animal Welfare Measure/Indicator	Scoring Description	Categorical Ranking in New Zealand Study
Stockpersonship animal-based measures on entering and exiting the race	Fearful/agitated Fall	% cows fearful/agitated in the race/forcing pen (climbing on others or attempting to escape) % cows lying in or falling in race/forcing pen or on exiting	0: 0.0% 1: 0.1–2% 2: >2%
	Stumble	% cows stumbling when exiting the race/holding pens into paddocks	0: 0–2.0% 1: 2.1–5% 2: >5%
	Run	% cows running out of the race/holding pens into paddocks	0: 0–5.0% 1: 5.1–10% 2: >10%
Animal handling stockpersonship and resource-based measures	Mis-catch (in chute/race)	% cows mis-caught with gates on any part of the body either in the race or chute head bale	0: no mis-catch 1: mis-catch ≤1% 2: mis-catch >1%
	Hitting	% of cows hit or poked with moving aids	0: no hitting 1: occasional/few hit 2: frequent hit/poke (>10% cows)
	Noise of handlers Noise of equipment/machinery Dogs noise around the yard	Evaluate noise of handlers, noise of equipment (race or chute gate) and machinery (generators etc.) and observe the presence and noise frequency of dogs around the yard	0: no noise/dogs 1: minor audible/occasional noise 2: unpleasantly/ persistent noisy handlers/equip/dogs
	Health checks	Frequency of health checks on cows during pregnancy	0: daily 1: once-twice/week 2: less than weekly
	Yard flow of cattle	Yard flow of cattle influenced by handling facilities design/quality	0: very effective cattle flow 1: effective but with flaws 2: difficult flow
	Yarding frequency	Frequency of yarding of cows per year	0: >4 times 1: 3–4 times 2: 0–2 times

**Table 5 animals-10-01592-t005:** Descriptive statistics (from 25 Waikato beef farms) for measures that were included in the final protocol and which were recorded as percentage of observed animals.

Welfare Principles	Measures	Mean (%)	Min (%)	Max (%)	Percentiles		
					25	50	75
Good Feeding	Thin cows	10.7	0	61	2.6	5.7	10.0
	Poor rumen fill	30.6	0	68	15.5	29.9	45.7
Good Environment	Short tail	4.2	0	21	0.6	3.0	6.0
	Dirtiness	21.3	4	50	10.7	20.6	29.4
	Watery faeces	39.6	15	87	24.0	35.7	48.5
Good Health	Swelling	0.7	0	5	0.0	0.0	1.1
	Hair loss	0.1	0	1	0.0	0.0	0.0
	Abrasion	0.1	0	2	0.0	0.0	0.0
	Lameness	2.7	0	12	0.5	1.5	3.6
	Blindness	0.4	0	4	0.0	0.0	0.0
	Ocular discharge	1.5	0	8	0.0	0.0	3.2
	Nasal discharge	1.2	0	13	0.0	0.0	1.3
	Accidental deaths	0.6	0	2	0.0	0.4	1.2
	Deaths from health	2.0	0	7	0.9	1.6	2.9
	Culling for health	1.2	0	6	0.0	0.8	2.0
Stockpersonship	Fearful/Agitate	2.7	0	7	1.3	2.3	4.1
	Fall/lie	0.9	0	8	0.0	0.0	0.8
	Stumble	1.6	0	21	0.0	0.0	1.7
	Run exit	13.0	0	51	2.6	7.8	15.1

**Table 6 animals-10-01592-t006:** Observed frequencies from the 25 Waikato beef farms for handling/stockpersonship categorical measures. See Table 4 for description of assessments used to create these figures.

Measure	Categories and Number of Farms in Each Category		
Mis-catch	No mis-catch	˂1% of cows mis-caught	>1% of cows
	18	4	3
Hitting	No hitting	Few cows hit	>10% hit (frequent hit)
	18	4	3
Noise of handlers	No noise	Minor audible noise	Noisy handlers
	4	18	3
Noise of Equipment/machinery	No noise	Minor audible noise	Very noisy
	9	6	10
Dogs noise around the yard	No dogs around yard	Quiet dogs	Noisy dogs
	7	8	10
Health checks	Daily inspection 11	Once or twice a week 9	Longer than once/week 5
Yarding frequency	>4 times/year 5	Between 3–4 times/year 20	Below 3 times/year 0
Yard design flow	Effective	Minor problems	Significant problems
	13	7	5

**Table 7 animals-10-01592-t007:** Normally distributed and log-transformed traits indicating multiples of the standard deviation with a 15% cut-off thresholds at the 25 Waikato beef farms showing thresholds of 50% of farms in “green” (good welfare), those in orange and 15% of farms in the “red” (poor welfare).

Welfare Principles	Measures	Mean	Orange Thresholds	Red Threshold (for the Bottom 15% Farms)	Thresholds Imposed by Categorisation for Poor Welfare Score (Score 2)	Ratio of Red Threshold Over the Imposed Categorisation Value
Feeding	* % thin cows	10.7	5.3	19.3	>10%	1.9
	% hungry cows	30.6	19.1	75.3	>50%	1.5
Environment	* Short tail	4.2	2.6	8.1	>2%	**4.1**
	Dirtiness	21.3	17.5	36.1	>20%	1.8
	Diarrhoea	39.6	35.6	58.6	>20%	**2.9**
Health	* Swelling	0.7	0.4	1.5	>2%	0.8
	* Hair loss	0.1	0.1	0.4	>2%	0.2
	* Abrasion	0.1	0.1	0.4	>2%	0.2
	* Lameness	2.7	1.7	5.0	>2%	**2.5**
	* Blindness	0.4	0.2	0.9	>2%	0.5
	* Ocular discharge	1.5	0.9	3.1	>2%	1.6
	* Nasal discharge	1.2	0.6	2.2	>2%	1.1
	Dystocia	2.6	1.8	4.9	>2%	**2.5**
	Mortality rate	3.9	3.3	6.4	>2%	**3.2**
Stockpersonship	Fearful/Agitated	2.7	2.2	4.9	>2%	**2.5**
	* Fall	0.9	0.5	1.8	>2%	0.9
	* Stumble	1.6	0.7	2.8	>5%	0.6
	* Run	13.0	7.7	24.4	>10%	**2.4**

*** Normally distributed measures. Figures emphasized in bold/red font are measures where the ratio of derived threshold: imposed threshold was >2.

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
