# Peer review of "Developing an Animal Welfare Assessment Protocol for Cows in Extensive Beef Cow-Calf Systems in New Zealand. Part 2: Categorisation and Scoring of Welfare Assessment Measures"

_animals, 2020, doi:10.3390/ani10091592_

Round 1

Reviewer 1 Report

26: Kaurivi (Part A) should be indicated as the method used (based originally on...)

91: the number of cattle needs to be mentioned here as well.  More detail of the size of the farms (animal numbers etc should be included to indicate the range and type of enterprises assessed.

100: Indicate the process of developing the scores. All how /how many authors? How was this modified by the relevant literature?

110: keeping this consistency needs further clarification.  It would depend on the incidence of the disease 

127: the '15%' rule needs some justification.

138: The numbers don't add up and this is not explained: 32 measures / only 23 are mentioned in Tables 2 - 4 / 36 are graphed  in Figures 1 to 3.

333: Reconcile this conclusion with the decision to keep these measures consistent.  Perhaps a conclusion that it should be based on incidence?

Nevertheless this paper is an important one in the progress towards a consistent systematic approach to assessing welfare in beef herds in New Zealand.

Author Response

Thank you very much for the thorough review!

Kind regards,

Baby Kaurivi

Reviewer 2 Report

Excellent second part. Anyway, I ask the authors to refer to my suggestions in the first part  (animals-904308). I dont't repeat them here, because they had no opportunity to answer to them in the short time between review part 1 and 2. New comments for part 2:

Line

Comment

14

If You have a cow -calf farm, it has to be a beef farm, so shouldn’t it be just a cow – calf farm?

39

Or categorization? (English or US?): whole txt.

56

US or GB? Whole txt.

61

Dito

74

The discrimination can also be influenced by national animal protection law since there is often a threshold level by law for accepted or not accepted. Switzerland has a lot of measurements in animal keeping in animal protection law (https://www.admin.ch/opc/de/classified-compilation/20080796/index.html)

111

Why have the authors not used threshold by national animal protection law or natural break by Jenks for categorisation?

384

The reader needs more information how the authors come to divide more or less arbitrary at the 15% and 5’0% percentile in god or poor animal keeping.

384

The authors have to discuss more in detail how they have dealed with short living parameters like rumen filling score and BCS? One is changing day by day and the other week by week. (same as in part 1).

384

At the beginning of part 2 the authors should give briefly the major results from part 1.

384

The author have to discuss the point what happens if one item is “red”. According to animal protection law in several countries, all items have to be fulfilled. In a lot of scientific publication only an overall score has to be satisfactory.

442

The authors mention only natural hazards from the terrain. What’s about artificial hazards from the farm building or infrastructure?

450

How can the herd be in “good health” when the mortality rate is high Differentiate and explain.

537

So a short-time parameter reflects the long-time animal keeping in New Zealand? That means animal keeping is very constant, which is very important and quite

Author Response

Thank you so much for your insightful review. Brilliant!

Kind regards,

Baby Kaurivi

Reviewer 3 Report

General comments

This study works towards filling an important gap in welfare assessment within pasture-based systems of livestock production. The study aimed to derive and test categorisation thresholds for poor, marginal or good welfare, for each of the measures outlined in part 1 of the study. Through this study the authors were able to attribute a 3 point score to each of the 32 measures described in part 1. They were then able to refine the thresholds using data obtained by applying the system to 25 commercial farms in NZ. Overall, this is an interesting and useful study which is presented well as a 2 part set of manuscripts.

This paper is once again largely well written and easy to read, however, I felt that some details were missing from the manuscript or required further clarification. As for my comments to the first part, I have made a lot of suggestions here but they are for the most part minor in nature and I hope they prove to be useful for the authors and not too onerous to address.

Introduction

A very nice introduction which appropriately covers the background and literature

44: [1].

47: [5-7] and (insert space)

63-64: check phrasing of end of sentence

78-79: Would need to rephrase this if you decide to move the summary data from Part 1 into part 2 as suggested

83: As for part 1, just ensure the aim is very clear and briefly state (one sentence or so) how your methodological approach would address those aims

Materials and methods

90: Refer readers to Tables 2-4 in the methods section, for details on how each of these measures were assessed

110: It is unclear what is meant by ‘kept constant’ here, please clarify

120: This essentially describes the analysis for the summary statistics presented in Part 1 (lines 102-103 in P1) in a more complete way than Part 1 had done. This would make it very easy to move that data from P1 to P2 as suggested! (if doing so also remember to remove lines 102-103 from the part 1 methods as they would not be relevant)

128: Perhaps also explain why 50%

Results – categorised thresholds

132: Section 3.2 and the summary statistics from part 1 could slot in here

133-135: This could be rephrased to be a little clearer. i.e. that Tables 2-4 show how you chose to categorise each of the variables (which may actually be better suited to the methods section rather than a result), then the figures indicate how the actual observed data from the 25 farms fell within those pre-determined categories.

152-153: This currently reads that these measures do not result in poor welfare in general, which is in contrast with their inclusion in the protocol – rephrase to clarify that the farms tested did not score poorly for these measures

153: 2 months, resulting in a poor welfare score

155: farms received a marginal welfare score

156:” of minimal welfare concern” – again implies these are not a problem in general rather than that they were not observed – please rephrase

157: maybe: dog noise positioned 10 and 11/25 in category 2

Tables 2-4

As noted above, I believe these tables belong within the methods section rather than the results section.

There are currently inconsistencies in how the tables are formatted and also in the information presented in the tables. Why is ‘welfare principles’ included for Table 2 but not the others?

The column names are inconsistent: Categorical ranking vs categorical ranking in New Zealand study.

I assume that the tables have been separated into 3 to avoid having a potentially overwhelmingly large table, but, by separating them out I find myself looking for a difference in the content or format of the tables that does not exist. I think the tables could therefore be combined into one large table, which I believe will be clearer and easier to interpret. Inclusion of the column “welfare principles” would still allow you to separate the measures into the categories of good feeding, appropriate environment, good health and appropriate stockpersonship, while keeping everything together. If you choose to keep these separate, not that the heading for Table 3 appears to have been accidentally copied over from Part 1 and needs to be corrected.

Check the middle categorical rankings are correct throughout tables 2-4 and remove any unnecessary “>” (i.e. either >250≤500m should be 250≤500m OR the 0 category should be 0-250m rather than 0<250m etc. – make sure this is consistent throughout the tables)

Table 2

Absence of hunger: “scale; [22])

Rumen fill: Use “:” instead of “-” after numerical categories for consistency

Dirty body: remove unnecessary “,” after flank and perhaps elaborate on definition of dirty here – how much coverage to be considered dirty?

Ease of movement: remove underline of text, specify minimal hazards and clarify what is meant by “(name)” or remove this

Table 3

Blind eye: specify blind in one or both eyes?

Disbudding castration: specify age at disbudding and castration are 2 separate measures?

Tagging/notching: specify no tag or use of anaesthetics regardless of tagging or notching procedure

Table 4

Phrasing of the welfare criteria are a little unclear i.e. “Stockpersonship animal-based measures in and out of race”

Fearful: specify where % cows fearful/agitated was recorded – race/holding pen?

Stumble: stumbling when exiting the race

Mis-catch: % cows mis-caught with gates

Noise: This is quite vague and subjective, perhaps you could give relatable examples of how much noise is considered minor vs too much?

Yarding frequency: As mentioned in my review of part 1, I am not aware of any evidence that sufficiently supports the thresholds attributed to yarding frequency here in the context of welfare. I believe it comes down to the quality of the handling and mustering more than the frequency and must disagree with its inclusion in the protocol.

Figures 1-3: Following from my comments on the table, this could perhaps be a single, 3-part figure.

The measures displayed within the figures are currently not consistent with the measures listed in the tables (e.g. vaginal prolapse was not mentioned previously?). I feel that these should correspond. It would also be more intuitive if the measures were also presented in the same order for the table and figures.

The figures headings are currently inconsistent and a little unclear. Perhaps rephrase all to “Frequency analysis of categorised [insert welfare principle/criteria e.g. good feeding] measures on the 25 Waikato beef farms, for which scores were assigned as either 0: good, 1: marginal, or 2: poor welfare. See Table X for further information on how each measure was categorised into a score of 0, 1 or 2.”.

Please adjust the formatting of the numbers on each of the bar charts that are currently overlapping and unable to be read (e.g. numbers in the categories 0 and 1 for faecal staining)

Figure 2. The ear tag, castration and disbud measures do not fit within this table as is – the scores of 0%, >0-2% and >2% are not applicable – please move these to another figure or reword the figure headings as suggested above.

Figure 4: Please clarify the layout of the figure in the figure heading. Note specifically that each farm is individually represented along the X axis or add this information as an axis title on the chart. This could also be more clearly explained in the text (Line 169).

Results – refined thresholds

178: perhaps reiterate here that the values 15% and 50% were derived from literature by including the relevant citations

181: Perhaps stick to poor, good or marginal thresholds or scores 0, 1 and 2 for consistency? This should be the same in Table 5 as well

Table 5: Check phrasing and grammar in table heading

Using two layers of table headers is a little confusing – you could remove the first row altogether here without losing information. Instead, indicate specifically which measures were normally distributed and which were log transformed using a symbol/footnote.

The use of * in the table and footnote is unclear and is not consistent with the bold/red font. Is the value 75.3 for % hungry cows supposed to be bold? – it currently appears to be

Perhaps rephrase the footnote to “Figures emphasised with bold/red font are…”

Perhaps only use the bold/red in the ratio column and not also the red threshold column

Discussion

The phrasing throughout the first few paragraphs of the discussion is often difficult to follow and requires clarifying.

190:193: Sentences appear to be repetitive – perhaps rephrase

195-6: ”average of 10.7% of cows having a BCS”

196-198: Please rephrase this sentence, it is currently very unclear. Also specify that it still refers to BCS.

199: remove “,” after “impaired”

200-201: Perhaps rephrase: Other studies have suggested that a threshold for the proportion of thin cows that is deemed unacceptable could be set at 5-15% [24] or 6.7% [16].

219: 20%

242: tomos).

273: [32]).

286-287: Vapocoolant is an anaesthetic not analgesia

295: warranty -> warrant

296-298: This sentence is unclear

296-302 and 307-209: This again appears to contradict your inclusion of yarding frequency as a measure of welfare and the thresholds you have proposed. I would again suggest that yarding frequency could be removed altogether from the protocol and from the two papers, or will require justification if retained in the protocol.

It would be useful to see some further discussion of the limitations of your approach with respect to your small sample size. Ongoing modification of the derived thresholds may be necessary if the performance of the farms within your study do not accurately represent cow-calf farming systems in NZ more broadly. Some further discussion around the next steps for validating this protocol and the need for additional expert opinion may also be useful to include within the discussion, with greater elaboration than was given in the conclusion.

It may also be useful to include somewhere (perhaps as an appendix?) the final protocol and categorisation thresholds that were determined throughout the study, using the mix of imposed and derived thresholds that you would like to see used in future studies and ultimately applied for welfare assessment. The ‘product’ here is not currently given in its final form.

“6.Patents” remove if not relevant

General comment: Overall I like the approach taken in this paper and again commend the authors for conducting a nice study that will be a useful contribution to the field of livestock welfare assessment.

Author Response

Thank you so much for the thorough, insightful review. I enjoyed working through your comments and suggestions. You are just brilliant at this!

Kind regards,

Baby Kaurivi
